# In Silico Designed Multi-Epitope Immunogen “Tpme-VAC/LGCM-2022” May Induce Both Cellular and Humoral Immunity against *Treponema pallidum* Infection

**DOI:** 10.3390/vaccines10071019

**Published:** 2022-06-25

**Authors:** Lucas Gabriel Rodrigues Gomes, Thaís Cristina Vilela Rodrigues, Arun Kumar Jaiswal, Roselane Gonçalves Santos, Rodrigo Bentes Kato, Debmalya Barh, Khalid J. Alzahrani, Hamsa Jameel Banjer, Siomar de Castro Soares, Vasco Azevedo, Sandeep Tiwari

**Affiliations:** 1Laboratory of Cellular and Molecular Genetics (LGCM), PG Program in Bioinformatics, Department of Genetics, Ecology, and Evolution, Institute of Biological Sciences, Federal University of Minas Gerais (UFMG), Belo Horizonte 31270-901, Brazil; lucasgabrielrgomes@ufmg.br (L.G.R.G.); thaiscristinavr@gmail.com (T.C.V.R.); arunjaiswal1411@gmail.com (A.K.J.); roselanegr@gmail.com (R.G.S.); rbkato@gmail.com (R.B.K.); dr.barh@gmail.com (D.B.); 2Centre for Genomics and Applied Gene Technology, Institute of Integrative Omics and Applied Biotechnology (IIOAB), Nonakuri, Purba Medinipur 721172, West Bengal, India; 3Department of Clinical Laboratories Sciences, College of Applied Medical Sciences, Taif University, P.O. Box 11099, Taif 21944, Saudi Arabia; ak.jamaan@tu.edu.sa (K.J.A.); h.banjer@tu.edu.sa (H.J.B.); 4Department of Immunology, Microbiology, and Parasitology, Institute of Biological and Natural Sciences, Federal University of Triângulo Mineiro (UFTM), Uberaba 38025-180, Brazil; siomars@gmail.com

**Keywords:** *Treponema pallidum*, sexually transmitted infection, syphilis, chimeric multi-epitope vaccine, immunoinformatics

## Abstract

Syphilis, a sexually transmitted infection caused by the spirochete *Treponema pallidum*, has seen a resurgence over the past years. *T. pallidum* is capable of early dissemination and immune evasion, and the disease continues to be a global healthcare burden. The purpose of this study was to design a multi-epitope immunogen through an immunoinformatics-based approach. Multi-epitope immunogens constitute carefully selected epitopes belonging to conserved and essential bacterial proteins. Several physico-chemical characteristics, such as antigenicity, allergenicity, and stability, were determined. Further, molecular docking and dynamics simulations were performed, ensuring binding affinity and stability between the immunogen and TLR-2. An in silico cloning was performed using the pET-28a(+) vector and codon adaptation for *E. coli*. Finally, an in silico immune simulation was performed. The in silico predictions obtained in this work indicate that this construct would be capable of inducing the requisite immune response to elicit protection against *T. pallidum*. Through this methodology we have designed a promising potential vaccine candidate for syphilis, namely Tpme-VAC/LGCM-2022. However, it is necessary to validate these findings in in vitro and in vivo assays.

## 1. Introduction

Syphilis is a sexually transmitted infection (STI) caused by *Treponema pallidum* subspecies *pallidum*, which belongs to the genus *Treponema*. Other pathogens from the genus cause various other non-venereal infections, such as endemic syphilis (*T. pallidum* subsp. *endemicum*), yaws (*T. pallidum* subsp. *pertenue*), and pinta (*T. carateum*) [1]. *T. pallidum* subsp. *pallidum* is a spiral-shaped, slow-growing, obligate human pathogen with a genome size of approximately 1.14 Mb, a protein count of 967, and a GC% content of 52.8%. Long term culture methods for cultivating *T. pallidum* have only recently been developed, and require co-culture with Sf1Ep cottontail rabbit epithelial cells [2]. The disease occurs in three stages: primary (First two to three weeks, chancres at site of infection), secondary (lesions and rashes throughout body, can take months to resolve), and tertiary (long term neural and cardiovascular complications), alternated with a latent stage (asymptomatic), which can occur at any point of the infection [3]. *Treponema pallidum* subsp. *pallidum* can also cause congenital syphilis in infants through vertical transmission, known as mother-to-child transmission (MTCT). The infection occurs during gestation and can lead to fatal infections in neonates [3].

Syphilis continues to be a worldwide healthcare burden. Even though the infection is easily identifiable and treatable, it is endemic in developing countries and on the rise within selected populations, specifically, men who have sex with men (MSM) and female sex workers (FSW), particularly within the HIV-positive population in developed countries [4]. According to the World Health Organization (WHO), six million people are infected with syphilis each year, and 2.5 million cases of yaws, bejel, and pinta are reported each year [3,5].

Currently, Africa, the Americas, and the Western Pacific are the regions most afflicted by incident cases [5]. Of those, an exceptional burden is on populations of pregnant women. Untreated syphilis in pregnancy is a leading cause of fetal and antenatal morbidity and mortality, resulting in high numbers of stillbirths, preterm infants, and cases of congenital syphilis [4]. According to the WHO, syphilis causes about 200,000 fetal and neonatal deaths each year, and about 215,000 infants are placed at risk of early death [5]. Gay men and sex workers are two other populations at-risk, with a high prevalence of STIs such as syphilis [5].

While syphilis can still be treated with penicillin with no occurrence of penicillin-resistant strains, there have been cases of macrolide-resistant strains, such as Azithromycin [6]. The disease’s prevalence in spite of the pathogen’s sensitivity to penicillin indicates the pathogen is unlikely to be controlled through screening and treatment alone [7]. Despite concerted efforts from the WHO to contain congenital syphilis and joint efforts to halt the spread of sexually transmitted syphilis, it remains a challenging disease to tackle as a result of various limiting factors [8]. Among said limiting factors, the pathogen’s difficulty to cultivate and study, the social stigma associated with the disease, its comorbidity with HIV, and the absence of a vaccine are of note [8].

Proof-of-principle for successful syphilis vaccination in the rabbit model was established in 1973, indicating development of a *T. pallidum* vaccine could be viable, but the immunization procedure demonstrated then was untenable for human application [9]. Attempts to develop a syphilis vaccine have since focused on the targeting of the bacterium’s outer membrane proteins (OMPs), of which there are few, and the few that are known are difficult to isolate [7,9]. Certain OMPs had limited potential in eliciting a protective response, suggesting that no single protein will confer full protective immunity against *T. pallidum* [10]. In this study, we sought to tackle the limitations to syphilis vaccine design through the use of computational biology and using an immunoinformatics-based approach.

Modern computational biology techniques, such as reverse vaccinology (RV) and immunoinformatics, have proven to be powerful tools for reducing resources and minimising the time typically spent in developing vaccines [11]. The usage of RV allows us to identify potential vaccine targets of interest in the pathogen’s genome while ensuring that no host-homologous sequences are in use. Immunoinformatics allows us to filter epitopes of the target protein according to their capacity to induce an immune response in the host [12]. It has been used to identify vaccine targets and drive vaccine development for a number of bacteria, viruses, parasites, fungi, as well as for cancers [13].

## 2. Materials and Methods

### 2.1. Selection of Target Antigenic Proteins

In order to search for and determine epitopes capable of eliciting an immune response from the host, 15 proteins, previously determined to be potential vaccine targets through reverse vaccinology, were selected from the work of Jaiswal et al. 2017 [14]. Their analysis of the Pan-genome of all available *T. pallidum* genomes identified potential vaccine targets in the core genome of *T. pallidum* sequences. Since these are core proteins, which are present in all *T. pallidum* strains, and were predicted to be surface-exposed, they are likely to have high expression levels across multiple strains and high immunogenic potential [14]. In addition, the proteins are non-homologous to the host, likely essential to the pathogen, and were predicted to be potential antigens [14]. Three other proteins considered to be immunogenic in previous in vivo studies using the outbred rabbit model were also added [15,16,17]. The amino-acid sequences for each protein, used as the database for this search, were retrieved from the National Center for Biotechnology Information (NCBI) Database. Eighteen proteins were selected for analysis (Appendix A).

### 2.2. Prediction of MHC-I Allele Binding CTL Epitopes

For the cytotoxic T lymphocyte (CTL) epitope prediction, in an effort to improve the confidence in the selected epitopes, two different platforms were used, both of which were used to predict nine amino-acid residue long sequences. The Immune Epitope Database and Analysis Resource (IEDB-AR) is robust and has multiple epitope prediction tools [18]. The major histocompatibility complex I (MHC-I) binding epitope prediction tool was used to identify CTL epitopes in the target proteins. In order to design an immunogen capable of inciting a response in a wide range of population worldwide, a reference set of 27 MHC-I alleles, which has a high frequency in the global population, was selected for epitope binding [19]. The IEDB-AR recommended 2020.09 (NetMHCpan EL 4.1) prediction method was used. Only epitopes having a percentile rank of <1% and an IC50 of <500 nM were selected.

We also applied a tool to assess antigenic processing and transportation. This tool, NETCTL-1.2, uses the ANN and SMM methods to perform predictions. All of the allele supertypes (A1, A2, A3, A24, A26, B7, B8, B44, and B58) [20], which were used in the previous step and are available in this platform, were also used in this step.

### 2.3. Prediction of MHC-II Allele Binding HTL Epitopes

To predict MHC-II epitopes for binding to Helper T Lymphocytes (HTL), we used two high-quality predictors [21] to identify the higher confidence epitopes, with a standard sequence length of fifteen amino acid residues. The IEDB-AR MHC-II binding epitope [18] and NETMHCII-2.3 [22] tools were used in combination. In this step, only IEDB-AR epitopes with a percentile rank < 3% and IC50 < 1000 nM were kept in the study. The platform NETMHCII-2.3 used an ANN with diverse epitope databases to perform the predictions. High-frequency alleles described in Greenbaum et al., 2011 [19] were used in the MHC-II prediction, both for the IEDB-AR and the NETMHCII 2.3 predictions.

### 2.4. Prediction of B-Cell Epitopes

We have used the ABCpred tool, an ANN-based tool for the prediction of linear B-cell epitopes, using the default parameters [23].

### 2.5. Filtering Best Epitopes from Each Protein

We run an in-house python script to determine shared epitopes between the two MHC-I and MHC-II prediction methods and select the predicted epitopes with high confidence between the prediction methods. Finally, we run the script to select overlapping epitopes between MHC-I and B epitopes and MHC-II and B epitopes, that is, epitopes capable of inducing cellular and humoral immune responses. In this case, the window was 2–9 residues for CTL epitopes and 2–15 residues for HTL epitopes. Only the epitopes that showed overlaps in the two cases were selected for further analysis. In order to filter down the number of epitopes and carefully define the final structure, further filtering steps were performed: MHC-II epitopes were filtered based on IC-50, keeping only epitopes with IC50 of up to 50 nM [24]. MHC-I epitopes were filtered using IEDB-AR’s immunogenicity tool with a cut-off of 0.1, indicating a higher probability of selecting immunogenic epitopes [25]. Only overlapping CTL and HTL epitopes were selected. The remaining epitopes that differed by only one or two residues were filtered, selecting only those with lower IC50 or higher immunogenicity. The final epitopes were used to construct two different chimeric proteins, which were compared in regards to the overall population coverage of the alleles used in their construction [26], as well as their physico-chemical properties.

### 2.6. Construction of Multi-Epitope Immunogen Sequence

To determine the final sequence of the chimeric protein, the final epitopes were merged using appropriate linker peptides, the purpose of which is to assist in protein folding and processing. CTL epitopes were linked by AAY linkers, and MHC-II epitopes were joined by GPGPG linkers [11]. The selected adjuvant was the cholera enterotoxin B-subunit (ctxB) [27], linked to the rest of the sequence by the peptide linker EAAAK. Two separate immunogens were constructed, and the immunogen with the higher overall population coverage was selected.

### 2.7. Prediction of Antigenicity, IFN-γ Induction, Toxicity, and Allergenicity of the Multi-Epitope Immunogen

The final chimeric protein structure was subjected to several analyses to answer important questions regarding its induction of immune response, allergic and toxic potential, and physico-chemical properties. First, VaxiJen was used to assess the antigenic capacity of the amino acid sequence through the automatic cross-covariance method, evaluating the physico-chemical properties of the protein and predicting its immunogenicity without performing alignments [28]. Epitopes capable of inducing IFN-γ production with consequent TCD4+ lymphocyte activation were identified with the IFNepitope predictor, which uses an SVM hybrid method based on protein motifs to perform prediction [29]. The protein sequence was then evaluated for toxic potential by submitting it to Toxinpred [30]. In addition, Allertop v.2.0 was used to evaluate the protein’s allergenic propensity based on the amino acid chain structure [31].

### 2.8. Physico-Chemical Properties and Host and Microbiota Homology Analyses

The molecular mass, theoretical pI, extinction coefficient, aliphatic index, grand average of hydropathicity (GRAVY), estimated half-life for three model organisms (Escherichia coli, yeast, and mammal cells), and instability index of the final protein sequence were evaluated through the ProtParam tool [32]. The solubility index was measured through the Protein-Sol tool [33], which evaluates the protein based on E. coli expression data. The Pipeline Builder for Identification of drug targets for infectious diseases (PBIT) tool [34] was used to search for homology between the chimeric protein and the proteome of the host as well as the gut microbiota of the host.

### 2.9. Secondary Structure Prediction

The PSIPRED prediction tool was used to determine the secondary structure of the final protein. This tool makes use of a complex ANN and Position-Specific Scoring Matrix (PSSM) based approach to predict the structure and generate pictures [35]. The RaptorX tool was used to provide the ratios of β-strands, ɑ-helixes, and coils.

### 2.10. Tertiary Structure and Refinement

We used Phyre2 intensive model, RaptorX, and I-TASSER servers to select the best structure for tertiary structure prediction. The Phyre2 intensive method consists of multiple alignments of the sequence of interest with homologous sequences, followed by secondary structure prediction with PSIPRED. Information from these two steps was combined to determine a hidden Markov model. A search for this model was performed in an HMM database of proteins with known structures, and the model with the best score was used to determine modelling and correct errors [36]. RaptorX, which uses multiple-template threading (MTT) and scoring methods to indicate the quality of the models [37]. Finally, I-TASSER constructed the model with an iterative method based on templates according to fragment assembly simulations with further refinement [38]. Methods for the refinement of amino acid side chains using light and aggressive relaxation were applied through the GalaxyRefiner tool to improve model quality by enhancing the local and global structure of the chimeric protein [39]. To check the quality of the refined structure, we used the PROCHECK tool, available in the SAVES server V6.0, to generate the Ramachandran plot, comparing the structure of the chimeric protein with the geometry of amino acid residues resulting from high-quality structures [40].

### 2.11. Prediction of Conformational B Cell Epitopes

Conformational epitopes are indispensable in stimulating immune responses. The refined structure was submitted to ElliPro to predict these discontinuous epitopes [41].

### 2.12. Molecular Docking between the Chimeric Protein and the TLR-2 Recepto

The Toll-like receptor 2 was identified as a vital receptor in detecting T. pallidum infection and assembling an effective immune response against the pathogen [42]. We retrieved the Toll-like receptor-2 (TLR-2) structure from RCSB: (PDB ID: 2z7x) database to determine the interactions of the chimeric protein with this receptor. The structure was edited with the Chimera visualization software, removing water molecules, ligands and side chains [43]. In order to verify the interactions between the chimeric protein and the TLR-2, molecular docking was performed using the Swarmdock server. The proteins were subjected to blind docking, attempting to find the lowest energy conformations across the whole protein. [44]. Hydrogen bonds and hydrophobic interactions were evaluated using the LigPlot^+^ program [45]. The PDBePISA tool was used to calculate the solvation free energy gain (Δ^i^G) of the final selected complex [46].

### 2.13. Molecular Dynamics Simulation of the Receptor-Ligand Complex

We performed the molecular dynamics simulation using the Gromacs v5.0 program [47] to enhance understanding of the microscopic structural properties of the interaction between the chimeric protein and the Toll-like receptor. To set simulation parameters, we prepared the software in the following manner: To construct protein topology and information about bonded and non-bonded characteristics, pdb2gmx will be used. The structure will be solvated in a cubic box of TIP3P water molecules. The complete system simulation was performed with the GROMOS96 43A1 force field, and a concentration of 150 mM sodium chloride (NaCl) ions were introduced to neutralize the system. Energy minimization was executed to ensure the quality of the system’s geometry and the absence of steric clashes. For this purpose, the steepest descent algorithm was applied. Simulation time was 90 ns.

### 2.14. In Silico Cloning

In silico cloning was performed to verify the capacity of cloning and expression of the protein in an appropriate expression vector. For this, the codon usage of our peptide sequence was adapted according to the codon usage of the *E. coli* expression system. For this purpose, the JCat tool was used for reverse translation. From that cDNA sequence, the codon optimisation for *E. coli* k12 was performed, returning the Codon Adaptation Index (CAI), which must have a score higher than 0.8, and the GC content should be between 30–70% [48]. Simulation and visualization of the in silico cloning were performed through the SnapGene^®^ software (from Insightful Science, available at snapgene.com) accessed on 1 October 2021, where the sequence of the chimeric protein was inserted in the pET-28a(+) plasmid with the help of Blpi and BamHI restriction enzymes.

### 2.15. Immune Simulation of Multi-Epitope Immunogen

We used the C-ImmSim server to run an immunological simulation to enhance the description of the immune response outlined by the chimeric protein [49]. The in silico method uses PSSM for epitope prediction and machine learning to assess interactions. The model also simulates the anatomical regions where crucial events of immunity occur: the bone marrow, where the lymphoid and myeloid cells are produced; the thymus, where the autoreactivity process happens; and the tertiary lymphatic organ, where antigenic presentation occurs, which describes the immunogenic profile. Three injections containing 1000 immunogen proteins each were given at four-week intervals for the simulation. Time steps were set at 1, 84, and 168 (each step representing eight real-life hours and time step 1 being injection time = 0). The total steps were modified to 1050, and other parameters were kept at the default. To check the effectiveness of the selected epitopes, we used the C-ImmSim tool again to simulate injections for only the adjuvant sequence while maintaining parameters, as described above.

## 3. Results

### 3.1. Predicted CTL Epitopes

Epitopes were predicted using the IEDB-AR database, yielding epitopes that can be recognized by MHC-I alleles with high frequency in the global population for all 18 proteins under analysis. We also submitted these proteins to the NETCTL 1.2 server to improve the confidence of the chosen epitopes. These epitopes had the size of nine amino acid residues.

### 3.2. Predicted HTL and B-Cell Epitopes

MHC-II epitopes were predicted through the IEDB-AR and netMHCII 2.3 tools. IEDB-AR epitopes were predicted to bind to the most common MHC-II alleles, according to the IEDB-AR database. These epitopes had a size of 15 amino acid residues.

The protein sequences were submitted to the ABCpred tool, and 16-mer epitopes, with scores higher than 0.51, were predicted for all 18 proteins according to the ability to interact with B lymphocyte receptors.

### 3.3. Overlapping Epitopes for Both Humoral and Cellular Responses

High confidence epitopes were selected based on their overlap between the two methods for each category, MHC-I and MHC-II, for each protein at a time. Finally, epitopes capable of inducing both humoral and cellular responses were selected according to the overlap between each category and B epitopes. The number of overlapping MHC-I and B epitopes was 729. The number of overlapping MHC-II and B epitopes was 521. These epitopes were subjected to the next screening methods. After the IC-50 screening, 112 MHC-II/B epitopes fit the new threshold. When the immunogenicity screening was applied, 269 MHC-I/B epitopes had scores higher than 0.1. The remaining epitopes were subjected to a search for overlapping MHC-I and MHC-II epitopes, keeping 37 MHC-I epitopes and presenting sequences overlapping with 32 MHC-II epitopes. These epitopes were used to construct two different chimeric immunogens, which were then compared in regards to their overall population coverage and physico-chemical properties. The chimeric protein that was selected was composed of 11 MHC-I epitopes and 15 MHC-II epitopes belonging to ten different proteins (Table 1). The final version of Tpme-VAC/LGCM-2022 had an overall population coverage of 99.93% of HLA Alleles (Appendix A).

### 3.4. Constructed Multi-Epitope Vaccine Sequence (Tpme-VAC/LGCM-2022), and Host and Microbiota Homology

The Treponema Pallidum Multi Epitope-Vaccine/Laboratory of Cellular and Molecular Genetics-2022 (Tpme-VAC/LGCM-2022) is composed of the following sequences: the cholera enterotoxin B subunit (ctxB) followed by the linker peptide EAAAK. Then, CTL epitopes linked by AAY linker peptides, followed by HTL epitopes linked by GPGPG linker peptides (Figure 1A). The chimeric protein was found to be non-host and non-gut-microbiota homologous.

### 3.5. Secondary and Tertiary Structural Properties of Tpme-VAC/LGCM-2022

The result of PSIPRED showed that, among the 558 residues of the sequence, there was an arrangement of 48% ɑ-helices, 15% β-strands, and 37% coil formation (Appendix A).

Three different prediction methods were applied to find the model with the best structural quality. The highest quality model was constructed by the RaptorX server, the Ramachandran plot of this model showing 92.9% of residues in most favored regions (Appendix A). After structural refinement through the GalaxyRefiner tool, the highest quality model had 94.9% amino acid residues in the most favored regions and only 0.2% in a disallowed region (Figure 1B,C).

### 3.6. Antigenicity, IFN-γ Production, and Conformational B-Cell Epitopes in Tpme-VAC/LGCM-2022

The chimeric protein sequence was determined as a probable antigen according to the VaxiJen tool, with a score of 0.6852. The IFNepitope tool, using the SVM method, predicted 230 epitopes with positive and negative scores. Among these, 69 epitopes had a score greater than 1, considered more capable of inducing the production of this cytokine (Appendix A). The final refined structure of the protein was submitted to ElliPro, and five conformational epitopes with scores above 0.7 were predicted (Appendix A).

### 3.7. Physico-Chemical Properties, Toxicity, and Allergenicity of Tpme-VAC/LGCM-2022

According to the ProtParam web tool, the theoretical molecular mass of the protein is 56,596.15 (56.59 Kd), and its isoelectric point (pI) is 9.03, indicating activity in a basic environment. The Instability Index (II), which is related to the stability of the protein, is 29.92, characterizing it as stable. Its estimated half-life in in vitro mammalian reticulocytes is 30 h, >20 h in in vivo yeast, and >10 h in in vivo *E. coli*. The protein’s aliphatic index, which is associated with stability in the face of temperature changes, was 83.89, the high indexes indicating greater stability. The grand average of hydropathicity (GRAVY) is 0.398, with positive values indicating hydrophobicity. According to Protein-Sol, the predicted scaled solubility was 0.378, a score that is lower than the solubility threshold, which is 0.45, related to solubility in E. coli, scores above it having higher solubility than the average. The pI according to this predictor was 9.520. According to the AllerTOP and ToxinPred tools, the chimeric protein sequence of the chimeric protein has shown no prospect of being allergic or toxic to humans.

### 3.8. Tpme-VAC/LGCM-2022docks with the TLR2 Receptor

Docking results were evaluated according to binding energy, number of members in a cluster, number of hydrogen bonds, and hydrophobic interactions. The binding energy between the chimeric protein with the TLR-2 receptor was −65.97, comprising ten hydrogen bonds. Three of the Tpme-VAC/LGCM-2022 residues involved in hydrogen bonds (Gln24, Thr22, His20P) belong to the adjuvant, while the remainder of the residues (Asp147, Tyr150, Ala185, Tyr189, Arg202) belong to the selected epitopes. These bonds were formed between the vaccine construct and the extracellular portion of the TLR-2 chain. Moreover, 21 residues were involved in hydrophobic interactions, of which 8 (Met1, Ile2, Lys5, Phe6, Gly7, Val8, Phe9 and Gly21) belonged to the adjuvant, while the remainder (Ala128, Leu135, Ala139, Val143, Leu149 Ala188, Phe192, Ala195, Ala199, Ala200, Ala203, Trp240 and Pro261) belong to the selected epitopes (Figure 2). PDBePISA calculated the Δ^i^G of the complex to be −20.6 kcal/mol. A negative Δ^i^G value corresponds to hydrophobic interfaces, or positive protein affinity.

### 3.9. Tpme-VAC/LGCM-2022-TLR2 Complex Is Stable in Molecular Dynamics Simulation

The stability of the interaction between the best-docked complex was evaluated by molecular dynamics using GROMACS 5.0. In MD simulation protocol, the energy minimized structure was carried out in phases: equilibration under a constant number of particles, volume, and temperature (NVT) at 300 K, and a constant number of particles, pressure, and temperature (NPT) at 1 bar, during which the protein atoms and the solvent molecules were allowed to equilibrate around the protein molecule for 1 ns. The system was analyzed using root-mean-square deviation (RMSD) and root-mean-square fluctuation (RMSF). The pressure plot indicated a fluctuation around 0.5 bar within the 1000 ps stabilization phase (Figure 3A). The temperature plot indicates the system maintained a temperature of 300 K during the same interval (Figure 3B).

The interaction was analyzed by RMSD, which reflects the complex’s structural stability. The RMSD plot shows a fluctuation ranging from 0.13 nm to 1.6 nm after an 80 ns time interval. This mild fluctuation indicates the stability of the complex during the tested time interval. To reflect the fluctuation of amino acid side chains, RMSF was analyzed. The RMSF plot shows mild fluctuations of RMSF values around 0.5, indicating uninterrupted interactions between receptor and ligand, and higher peaks with RMSF values around 2.0, indicating highly flexible loop regions in the complex (Figure 3C,D).

### 3.10. Codon Adaptation and in silico Cloning of Tpme-VAC/LGCM-2022

Using the JCat tool to perform *E. coli* k12 codon adaptation of the protein sequence, we obtained the reverse translated sequence for the protein. The GC content of the sequence was 56.45%, within the optimum range of 30–70%. The CAI index was 1.0 and within the allowed range. Using the SnapGene tool, we created *BlpI* and *BamHI* restriction sites to insert our Tpme-VAC/LGCM-2022 sequence into the pET-28a(+) plasmid vector. The complete length of the insert was 1680 bp (Figure 4).

### 3.11. Tpme-VAC/LGCM-2022 Could Simulate Immune Response

The results provided by the C-ImmSim tool for immune response simulation were compatible with and indicative of the development of immunity. Regarding the B cells population, the simulation predicted an increase in memory cells throughout immunogen injection points with a strong differentiation to the production of IgG and IgM and a decrease in memory B cells. Further, a significant production of the IgM + IgG, IgG1, and IgG1 + IgG2 immunoglobulins is noted over the injections, which are relevant to complement fixation, induction of innate response, and Th1 cell activation (Figure 5A–C).

In regards to T lymphocytes, an increase in T-helper populations with a strong Th1 differentiation was observed. Th1 lymphocytes are major inducers of cytotoxic T lymphocyte proliferation. They increase their cytotoxic capacity and stimulate the production of IFN-γ. The results also indicate a growth in the active cytotoxic cell population, with a decrease in the number of resting cells (Figure 5D–G). In regards to the innate immune system, the results followed the expected response patterns for the activation and proliferation of natural killer cells and macrophages.

In general, the immune simulation showed an increase, mainly during the secondary and tertiary response in cell types and cytokines that are vital in sustaining effective immune responses with clear peaks on injection days. There is also a clear decrease in the level of active cells a few days after the third injection, which, together with the evidence of the probable induction of the IL-10 and TGF-β cytokines, indicates an attempt by the immune system to control the response and prevent exacerbation (Figure 5H).

We compared the results from the simulation performed with those from our chimeric protein simulation using only the adjuvant. The comparison revealed a less intense proliferation of B cells populations on the adjuvant simulation, indicating the effect of the selected epitopes on the induction of B lymphocytes in silico. This pattern was not maintained in relation to immunoglobulin production, indicating a better activation of the humoral system by the adjuvant than the immunogen. The comparison of the simulations also showed similar patterns for the induction of cytotoxic T cells and cytokine production (Appendix A). A simulation performed with only the epitopes showed similar patterns of immune activation to the full immunogen simulation (Appendix A), indicating that the epitopes are responsible for the immune response elicited by the multi-epitope immunogen.

## 4. Discussion

The application of current immunoinformatics approaches in the development of multi-epitope immunogens has assisted in the acceleration of the slow and costly vaccine development process, as it allows for the efficient screening of genome sets and identification of candidates [13,50]. This class of vaccine is designed to trigger both the innate and adaptive immune systems, generate protective memory, and reduce the chance of side effects, as well as spontaneous reversions that occur in attenuated vaccines [50]. Peptide-based vaccines are also cheaper and easier to produce on a large scale, can be easily stored and transported due to freeze-drying, and do not require the cultivation of the infectious bacterium [50]. The limitations associated with *T. pallidum* research have meant that there is currently no clinical vaccine for syphilis. Some putative vaccine targets have been identified, but so far have not been successful in eliciting an adequate protective response in vivo [7].

The constructed immunogen may be used as a peptide-based subunit vaccine, and our in silico predictions indicate that it may be a strong candidate for protection against *T. pallidum*. In addition, the immunogen may also be used in the research and development of new diagnostic methods for syphilis [51].

The immunoinformatics strategy adopted in this study aimed to provide a multi-epitope immunogen consisting of epitopes selected from the core genome of the pathogen, thus ensuring its coverage of various strains and the scope of the displayed antigens. These epitopes were derived from eighteen *T. pallidum* vaccine targets. The *T. pallidum* Outer Membrane Proteome (OMPome) are generally regarded as promising candidates for vaccine targets in syphilis research [52]. Among the vaccine candidates selected for the study, four proteins (TP0897, TP0126, TP0326 and TP0733) are part of the *T. pallidum* OMPome [52].

During syphilis infection, the resolution of primary and secondary syphilitic lesions has generally been associated with cellular infiltrates of predominantly T-cells and macrophages. Clearance of primary lesions in humans is associated with a strong CD4^+^ T-cell response, as well as the presence of macrophages and NK cells, whilst clearance of secondary legions is associated with a higher abundance of CD8^+^ T-cells. The induction of delayed-type hypersensitivity (DTH) and Th1-mediated opsonophagocitosis, which is induced mainly by CD4^+^ T-cells, is thought to be the main mechanism of clearance in syphilitic lesions. While this evidence points strongly to the importance of cell-mediated immune response in the response to *T. pallidum* infection, there is also evidence pointing that the humoral immune response is also essential. *T. pallidum* opsonophagocitosis has been shown to be dependent on the presence of Immune Patient Serum, highlighting the importance of the production of opsonic antibodies in the response to syphilis [10].

Aiming to design an immunogen capable of eliciting the desired response against the bacterium, epitopes were selected for their ability to induce CTL, HTL, and B-cell responses in various HLA supertypes, to stimulate both cellular and humoral immunity in a broad range of the global population. This careful selection of epitopes also allowed for the exclusion of potentially deleterious sequences for the construct, resulting in a multi-epitope immunogen that is non-allergenic, non-toxic, and non-homologous to any host proteins. One disadvantage of this approach is that, in contrast with whole pathogen vaccines, multi-epitope immunogens lack some antigenic determinants, requiring the addition of an enhancer as an adjuvant with strong antigenic properties [27]. The cholera enterotoxin B-subunit (ctxB) protein was selected for this purpose.

The chimeric protein’s physico-chemical properties were determined to be within the acceptable parameters for recombinant protein production and application as a sub-unit vaccine. The sequence was optimized for expression in the *E. coli* K12 expression model and cloned into an *E. coli* expression plasmid. Through modelling, molecular docking, and molecular dynamics simulations, the multi-epitope immunogen was determined to be able to form a strong, considerably stable binding to the TLR-2. TLR-2 is one of the main Toll-like receptors involved in *T. pallidum* detection by the host, indicating its capacity to activate the receptor. To enhance our understanding of the protein’s immunogenic capacity, an immune simulation was performed. The results indicated it to be a putative inducer of both the innate and adaptive immune systems, with wide differentiation of B-cell population, antibody production, and high activation of Th1 Helper T-cells. Both types of responses are vital to the clearance of active treponemes from syphilis wounds and limit the bacterium’s ability to spread within the host [7]. The immunogen also induced both the differentiation of immune memory-associated cells and the production of important cytokines to balance the immune response.

## 5. Conclusions

In this study, a novel multi-epitope immunogen (Tpme-VAC/LGCM-2022), comprising high-ranked epitopes from eighteen *Treponema pallidum* proteins, was constructed using an immunoinformatics-based approach. Several criteria were applied to select epitopes capable of inducing strong humoral and cellular responses while assessing the protein’s allergenicity and toxicity to construct a safe immunogen. The designed immunogen has suitable structural, physiochemical, and immunological properties that can successfully elicit humoral and cellular immune responses against *T. pallidum*. Finally, the in silico immune simulation enhanced our understanding of the capacity of the immunogen to elicit an immune response, highlighting that it is capable of inducing both humoral and cellular immune responses. This feature is vital in the development of a syphilis vaccine, as both the clearance of active treponemes in infectious sites and prevention of its ability to spread within the host are key for effective syphilis prevention. Therefore, a promising immunogen candidate for *T. pallidum* was designed, notwithstanding, to ensure immunologic efficiency and memory development. Experimental validation must, however, be performed in vitro and in vivo. Further validation of Tpme-VAC/LGCM-2022 may be performed through the recombinant synthesis of the immunogen construct in a vector expression system (pET-28a(+), for expression in *E. coli*). Once synthesized and purified, the construct can then be used in in vitro and in vivo immunization assays in the rabbit model before progressing to human studies.

## Figures and Tables

**Figure 1 vaccines-10-01019-f001:**
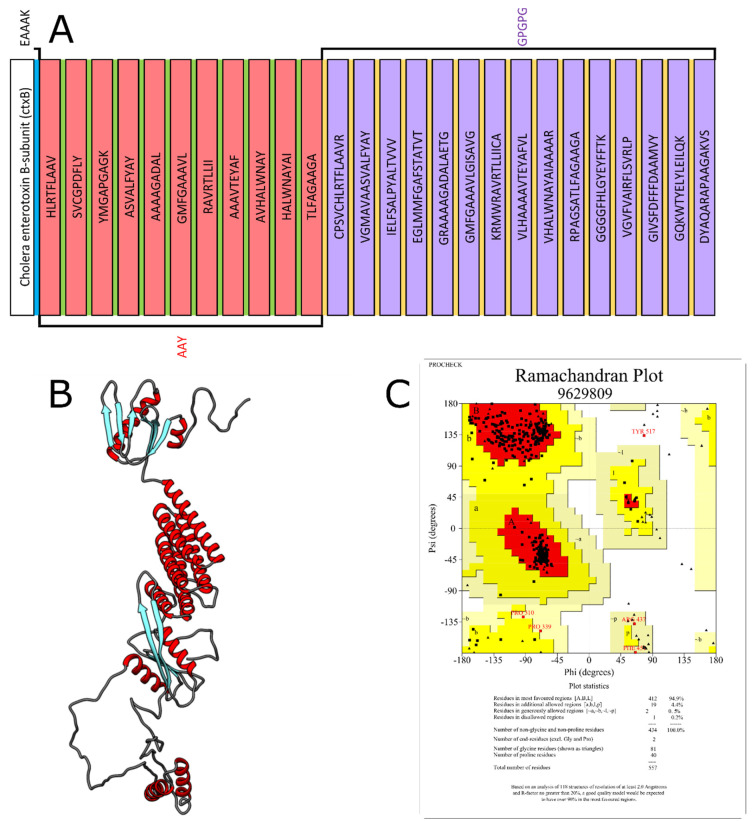
(**A**) Multi-epitope immunogen (Tpme-VAC/LGCM-2022) construct with highlighted peptide linkers and epitopes. Sequence length is 558 amino acid residues. (**B**) Three-dimensional structure modelling of the chimeric protein after refinement by GalaxyRefiner. (**C**) Ramachandran plot for the model after refinement, showing 94.9% residues in most favored regions and 0.2% in disallowed regions.

**Figure 2 vaccines-10-01019-f002:**
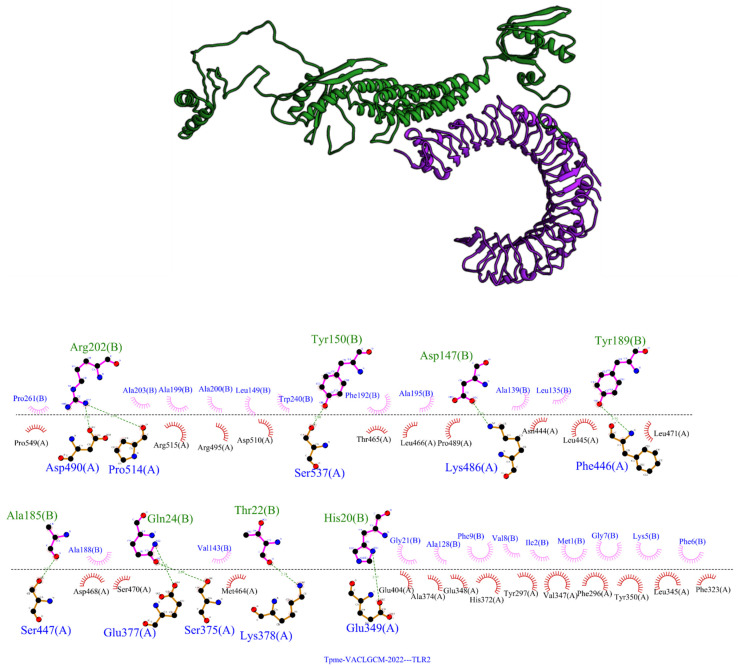
Molecular docking of the Tpme-VAC/LGCM-2022 with the TLR-2 receptor structure representing the interaction with the lowest energy score. 3D structure of the complex. In green, the chimeric protein, and purple is the TLR-2. 3D structure of the complex, highlighting 2D representation of the interactions between TLR 2 and the chimeric protein. The figure represents the residues of the chimeric protein (Chain-B) and TLR2 receptor (Chain-A) with hydrogen bonds (green dotted lines). The residues involved in hydrophobic interactions are shown (Red).

**Figure 3 vaccines-10-01019-f003:**
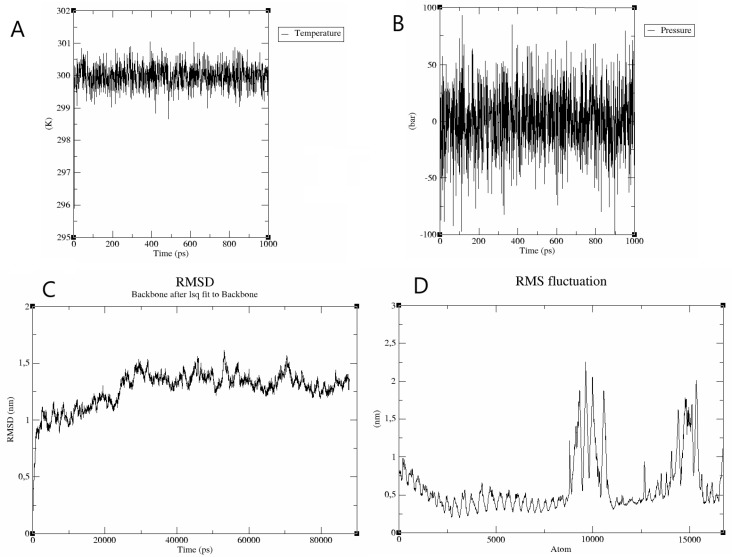
Molecular dynamics simulation plots for the Tpme-VAC/LGCM-2022-TLR2 complex. (**A**) The temperature plot shows that the temperature of the system reaches over 300 K, and fluctuates around 300 K throughout the equilibration phase (1000 ps). (**B**) The pressure plot displays pressure fluctuation during the equilibration phase (1000 ps), with an average pressure of 0.5 bar. (**C**) RMSD plot of the receptor-ligand complex shows no significant deviation, indicating a stable interaction. (**D**) RMSF plot shows mild fluctuations of about 0.5 nm and higher peaks with an RMSF value of 2, due to the highly flexible loop regions in the complex.

**Figure 4 vaccines-10-01019-f004:**
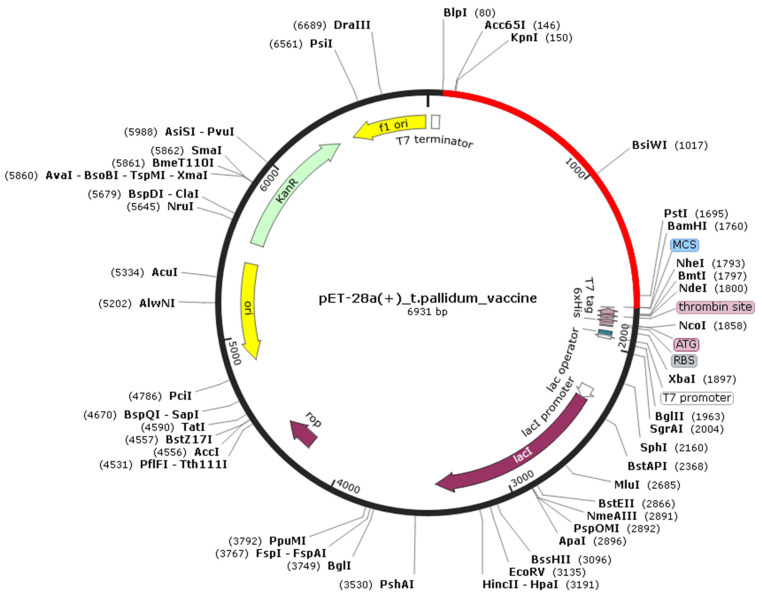
In silico cloning. Reverse translated sequence of the Tpme-VAC/LGCM-2022 protein insert represented in red between the *BlpI* and *BamHI* restriction sites. Vector is represented in black.

**Figure 5 vaccines-10-01019-f005:**
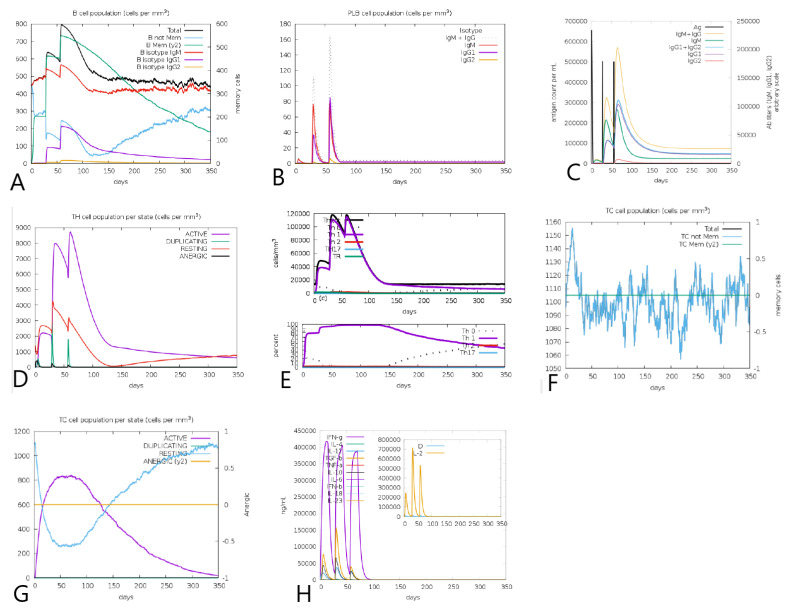
Immuno simulation results of the Tpme-VAC/LGCM-2022 regarding: (**A**) B cells population per mm^3^ (**B**) PLB cell population per mm^3^. (**C**) Immunoglobulin production. (**D**) Helper T-cell population per state. (**E**) I Helper T-cell differentiation. (**F**) Cytotoxic T-cell population. (**G**) Cytotoxic T-cell population per state. (**H**) Cytokine production.

**Table 1 vaccines-10-01019-t001:** Final epitopes selected for immunogen construct. Highlighted in bold is the overlap between MHC-I and MHC-II epitopes.

	GENE ID/NAME	MHC	EPITOPE	PERCENTILE RANK
**1**	TP_0049	I	HLRTFLAAV	0.12
**2**		II	CPSVC**HLRTFLAAV**R	0.9
**3**		I	SVCGPDFLY	0.22
**4**	TP_0323	I	ASVALFYAY	0.1
**5**		II	VGMAVA**ASVALFYAY**	1.1
**6**		II	IELFSALPYALTVVV	0.6
**7**		II	EGLMMFGAFSTATVT	0.7
**8**	TP_0335	I	AAAVTEYAF	0.14
**9**		II	VLHA**AAAVTEYAF**VL	0.8
**10**		I	AVHALWNAY	0.05
**11**		I	HALWNAYAI	0.21
**12**		II	V**HALWNAYAI**AAAAR	0.25
**13**		I	TLFAGAAGA	0.07
**14**		II	RPAGSA**TLFAGAAGA**	0.9
**15**	TP_0430/ntpK	I	AAAAGADAL	0.59
**16**		II	GR**AAAAGADAL**AETG	0.25
**17**		I	GMFGAAAVL	0.15
**18**		II	**GMFGAAAVL**GISAVG	0.4
**19**	TP_0435/nlpE	I	YMGAPGAGK	0.11
**20**	TP_0557	I	RAVRTLLII	0.72
**21**		II	KRMW**RAVRTLLII**CA	0.5
**22**	TP_0733	II	GGGGFHLGYEYFFTK	0.3
**23**	TP_0972/ftr1	II	VGVFVAIRFLSVRLP	0.12
**24**	TP_0326/BamA	II	GIVSFDFFFDAAMVY	0.12
**25**		II	GQKWTYELYLEILQK	0.03
**26**	tprK	II	DYAQARAPAAGAKVS	1.1

## Data Availability

Publicly available datasets were analyzed in this study. This data can be found here: https://www.ncbi.nlm.nih.gov/protein/ accessed on 1 May 2021.

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
