# Peer review of "In Silico Designed Multi-Epitope Immunogen “Tpme-VAC/LGCM-2022” May Induce Both Cellular and Humoral Immunity against Treponema pallidum Infection"

_vaccines, 2022, doi:10.3390/vaccines10071019_

Round 1

Reviewer 1 Report

This work provided a new perspective on facilitating the vaccine development of Syphilis using informatics and computation tools. It's a novel idea with excellent research potential, and also falls into the scope of the journal. However, it lacks some necessary validation steps. I would recommend a couple of revisions to proceed with publication.

1. Numerous computation tools, which decyphered correlation between 1-D amino acid sequence and the 4-D protein interactions with great accuracy, have emerged recently. Could you briefly introduce the mechanism of the tools used in this work and explain why they are the optimal tools for this work? 

2. The efficacy of the immunogen was only indirectly evaluated by simulations of its chemical or physical properties. To validate the outcome, could you demonstrate the efficacy directly through an in vitro study?

3. Please update the figures with high-resolution images. It's challenging to get all the details in the blurred figures.

Author Response

Dear Editor and Reviewers,

Firstly, we would like to thank the editor and the reviewers for reviewing the manuscript and providing us with valuable comments and suggestions. We are now submitting the revised version of the manuscript entitled “In silico designed multi-epitope immunogen “Tpme-VAC/LGCM-2022” may induce both cellular and hu-moral immunity against Treponema pallidum infection”, for publication in MDPI Vaccine (Manuscript ID: vaccines-1763576). Here, we have tried to the best of our knowledge to follow an appropriate and accurate way to do the corrections and respond properly. An explanation for each comment is given below.

Please find the answers to the comments and suggestions.

Thank you very much for your attention.

Sincerely,

Corresponding authors

This work provided a new perspective on facilitating the vaccine development of Syphilis using informatics and computation tools. It's a novel idea with excellent research potential, and also falls into the scope of the journal. However, it lacks some necessary validation steps. I would recommend a couple of revisions to proceed with publication.

  1. Numerous computation tools, which decyphered correlation between 1-D amino acid sequence and the 4-D protein interactions with great accuracy, have emerged recently. Could you briefly introduce the mechanism of the tools used in this work and explain why they are the optimal tools for this work? 

Reply: We would like to thank the reviewer for suggestions. We agree with the suggestion of reviewer. We used Phyre2 intensive model, RaptorX, I-TASSER servers and currently well-known tool AlphaFold, for the prediction of tertiary structure. Furthermore, best structures were selected based on each tool`s parameters ex. I-TASSER servers: C-score, AlphaFold: pLDDT (predicted lDDT-Cα), Phyre2: confidence >90%. Also, PROCHECK tool, available in the SAVES server V6.0 used to evaluate and select the high-quality tertiary structure followed by the refinement of structure using GalaxyWeb server.  In this work the RaptorX server has given the best tertiary structure compared to other tools.

  1. The efficacy of the immunogen was only indirectly evaluated by simulations of its chemical or physical properties. To validate the outcome, could you demonstrate the efficacy directly through an in vitro study?

Reply: We agree with the reviewer in that an in vitro study validating the efficacy of the immunogen against syphilis infection would be valuable for this research. However, due to the current global economic crisis, we have been unable to procure funding for such an experiment for now.

  1. Please update the figures with high-resolution images. It's challenging to get all the details in the blurred figures.

Reply: The resolution of images has been upscaled for better quality in the revised manuscript.

Reviewer 2 Report

This manuscript entitled, “In silico designed multi-epitope immunogen “Tpme-VAC/LGCM-2022” may induce both cellular and humoral immunity against Treponema pallidum infection” showed preliminary results of molecular docking and several criteria for inducing strong humoral and cellular responses, it will be useful study for further investigation. Author should be addressed in abstract and conclusion details about specific criteria used with conclusive statement. 

Author Response

Dear Editor and Reviewers,

Firstly, we would like to thank the editor and the reviewers for reviewing the manuscript and providing us with valuable comments and suggestions. We are now submitting the revised version of the manuscript entitled “In silico designed multi-epitope immunogen “Tpme-VAC/LGCM-2022” may induce both cellular and hu-moral immunity against Treponema pallidum infection”, for publication in MDPI Vaccine (Manuscript ID: vaccines-1763576). Here, we have tried to the best of our knowledge to follow an appropriate and accurate way to do the corrections and respond properly. An explanation for each comment is given below.

Please find the answers to the comments and suggestions.

Thank you very much for your attention.

Sincerely,

Corresponding authors

Comments and Suggestions for Authors

This manuscript entitled, “In silico designed multi-epitope immunogen “Tpme-VAC/LGCM-2022” may induce both cellular and humoral immunity against Treponema pallidum infection” showed preliminary results of molecular docking and several criteria for inducing strong humoral and cellular responses, it will be useful study for further investigation. Author should be addressed in abstract and conclusion details about specific criteria used with conclusive statement. 

Reply: We would like to thank the reviewer for the suggestions. More details have been provided in the abstract and conclusion to highlight and improve our conclusive statements.

Reviewer 3 Report

The authors aim to provide insights into the use of predictive bioinformatics methods for the potential development of multi-epitope peptide-based vaccines against Treponema pallidum infection. This manuscript is well written and clearly described with appropriate methodical implementation. Findings from this study are important and useful to readers, particularly in the field of vaccine development, I have some minor comments as provided below.

1. While I observed that only linear (continuous) B-cell epitopes were considered for B-cell epitope prediction, authors should explain/justify their non-usage of non-linear (discontinuous) B-cell epitopes in their vaccine construct giving the importance of these sets of epitopes to map out structural-functional hotspots on the target proteins.

2. Since the intention of the study is to design a potential globally-effective vaccine, authors should also emphasize their rationale for selecting MHC-II alleles for HTL-binding epitopes.

3. For the docking calculation, how did the author define/determine the interaction site for both proteins? and additionally, information on the interaction site (residues) should be clearly described as it is important to achieve target specificity in future studies.

4. Also, the docking score adopted to measure the affinity of the chimeric protein and TLR-2 may be less accurate since both systems are static in the docking experiment. Authors should complementarily determine the protein-protein affinity using ΔG energy estimations.

Author Response

Dear Editor and Reviewers,

Firstly, we would like to thank the editor and the reviewers for reviewing the manuscript and providing us with valuable comments and suggestions. We are now submitting the revised version of the manuscript entitled “In silico designed multi-epitope immunogen “Tpme-VAC/LGCM-2022” may induce both cellular and hu-moral immunity against Treponema pallidum infection”, for publication in MDPI Vaccine (Manuscript ID: vaccines-1763576). Here, we have tried to the best of our knowledge to follow an appropriate and accurate way to do the corrections and respond properly. An explanation for each comment is given below.

Please find the answers to the comments and suggestions.

Thank you very much for your attention.

Sincerely,

Corresponding authors

Comments and Suggestions for Authors

The authors aim to provide insights into the use of predictive bioinformatics methods for the potential development of multi-epitope peptide-based vaccines against Treponema pallidum infection. This manuscript is well written and clearly described with appropriate methodical implementation. Findings from this study are important and useful to readers, particularly in the field of vaccine development, I have some minor comments as provided below.

  1. While I observed that only linear (continuous) B-cell epitopes were considered for B-cell epitope prediction, authors should explain/justify their non-usage of non-linear (discontinuous) B-cell epitopes in their vaccine construct giving the importance of these sets of epitopes to map out structural-functional hotspots on the target proteins.

Reply: We would like to thank the reviewer for the valuable suggestion. Because the recognition of non-linear epitopes by the host immune system is mediated by these epitopes’ localization within the protein’s architecture, we elected instead to perform discontinuous B-cell epitope prediction on the final vaccine construct structure to validate whether or not the construct is capable of eliciting that response. The relevant discontinuous epitopes on the structure of Tpme-VAC/LGCM-2022 have been shown in the supplementary materials, Supplementary Table 3 and Supplementary Figure 4.

  1. Since the intention of the study is to design a potential globally effective vaccine, authors should also emphasize their rationale for selecting MHC-II alleles for HTL-binding epitopes.

Reply: We thank the reviewer for the comment and have amended the relevant section to make the rationale behind selecting these MHC-II alleles clearer in the text. Additionally, we evaluated the global population coverage of Tpme-VAC/LGCM-2022, that information can be found in Supplementary Figure 1

  1. For the docking calculation, how did the author define/determine the interaction site for both proteins? and additionally, information on the interaction site (residues) should be clearly described as it is important to achieve target specificity in future studies.

Reply: We thank the reviewer for the enlightening comment. The protein was docked blind to the TLR-2 (PDB ID: 2z7x) structure and the lowest binding energy conformation was selected. Three of the Tpme-VAC/LGCM-2022 residues involved in hydrogen bonds (Gln24, Thr22, His20P) belong to the adjuvant, while the remainder of the residues (Asp147, Tyr150, Ala185, Tyr189, Arg202) belong to the selected epitopes. Moreover, 22 21 residues were involved in the hydrophobic interactions, of which 8 (Met1, Ile2, Lys5, Phe6, Gly7, Val8, Phe9 and Gly21) belonged to the adjuvant, while the remainder (Ala128, Leu135, Ala139, Val143, Leu149 Ala188, Phe192, Ala195, Ala199, Ala200, Ala203, Trp240 and Pro261) belong to the selected epitopes. These bonds were formed between the vaccine construct and the extracellular portion of the TLR-2 chain.

  1. Also, the docking score adopted to measure the affinity of the chimeric protein and TLR-2 may be less accurate since both systems are static in the docking experiment. Authors should complementarily determine the protein-protein affinity using ΔG energy estimations.

Reply: We would like to thank the reviewer for the valuable suggestion. In the revised version of the manuscript, we have used PDBePISA to calculate ΔiG, or solvation free energy gain, upon formation of the interface, which is is a measure of the interface stability in protein complexes. ΔiG was found to be negative, which indicates positive protein affinity.